# Prognostic Potential of Galectin-9 mRNA Expression in Chronic Lymphocytic Leukemia

**DOI:** 10.3390/cancers15225370

**Published:** 2023-11-11

**Authors:** Agnieszka Bojarska-Junak, Wioleta Kowalska, Sylwia Chocholska, Agata Szymańska, Waldemar Tomczak, Michał Konrad Zarobkiewicz, Jacek Roliński

**Affiliations:** 1Department of Clinical Immunology, Medical University of Lublin, 20-093 Lublin, Poland; wioleta.kowalska@umlub.pl (W.K.); agata.szymanska@umlub.pl (A.S.); michal.zarobkiewicz@umlub.pl (M.K.Z.); jacek.rolinski@umlub.pl (J.R.); 2Department of Haematooncology and Bone Marrow Transplantation, Medical University of Lublin, 20-080 Lublin, Poland; sylwia.chocholska@umlub.pl (S.C.); waldemar.tomczak@umlub.pl (W.T.)

**Keywords:** CLL, Galectin-9, EBV, prognostic markers, biomarker

## Abstract

**Simple Summary:**

There are still very few studies on Galectin-9 (Gal-9) in chronic lymphocytic leukemia (CLL), and the effect of Gal-9 on CLL pathogenesis. This study considers the possible role of Gal-9 as a new prognostic biomarker in CLL. Gal-9 mRNA expression was quantified with RT-qPCR in purified B-lymphocytes of 100 CLL patients and analyzed in the context of clinical data. Our results revealed the upregulation of Gal-9 mRNA in malignant B-cells. Unfavorable prognostic markers were closely linked to high Gal-9 mRNA expression. An increase in Gal-9 mRNA expression was also correlated with a shorter time to treatment. Gal-9 plays a crucial role in the tumor microenvironment and may become a significant complement to other prognostic indicators. Furthermore, we propose that EBV coinfection may have a detrimental effect on the prognosis of CLL patients, partly due to Gal-9 expression upregulation caused by EBV.

**Abstract:**

Galectin-9 (Gal-9), very poorly characterized in chronic lymphocytic leukemia (CLL), was chosen in our study to examine its potential role as a CLL biomarker. The relation of Gal-9 expression in malignant B-cells and other routinely measured CLL markers, as well as its clinical relevance are poorly understood. Gal-9 mRNA expression was quantified with RT-qPCR in purified CD19+ B-cells of 100 CLL patients and analyzed in the context of existing clinical data. Our results revealed the upregulation of Gal-9 mRNA in CLL cells. High Gal-9 mRNA expression was closely associated with unfavorable prognostic markers. In addition, Gal-9 expression in leukemic cells was significantly elevated in CLL patients who did not respond to the first-line therapy compared to those who did respond. This suggests its potential predictive value. Importantly, Gal-9 was an independent predictor for the time to treatment parameters. Thus, we can suggest an adverse role of Gal-9 expression in CLL. Interestingly, it is possible that Gal-9 expression is induced in B-cells by EBV infection, so we determined the patients’ EBV status. Our suggestion is that EBV coinfection could worsen prognosis in CLL, partly due to Gal-9 expression upregulation caused by EBV.

## 1. Introduction

Galectins are a family of β-galactoside-binding lectins with a conserved carbohydrate recognition domain (CRD) [1,2]. Since their identification in 1976, 16 galectins have been described [3]. One of them, galectin-9 (Gal-9) has received special attention as a multifaceted player in adaptive and innate immunity, especially in T-lymphocyte development and homeostasis [4,5,6]. Gal-9 is an essential inhibitor of the immune system [5]. It promotes the development of regulatory T-cells (Tregs) and inhibits the differentiation of Th17 cells [3,5,7]. In addition, Gal-9 has the ability to trigger the apoptosis of Th1 cells and CD8+ cytotoxic T-cells via T-cell immunoglobulin and mucin domain 3 (TIM-3), resulting in a suppression of excessive inflammation [3,5,7]. However, apart from TIM-3, Gal-9 also binds to other receptors, including 4-1BB, Dectin-1, CD40 and CD44, on various immune cell subsets [3,5,8,9]. Nevertheless, TIM-3 is the best-known Gal-9-binding receptor. The interaction of Gal-9 and TIM-3 attenuates anti-tumor immunity and promotes tumor development [5,7,8,10]. Gal-9 is found in various tissues and cell types. It is highly expressed in bone marrow and lymphoid tissues [5]. Gal-9 expression is found not only in normal cells and tissues but also in pathological conditions, both in solid tumors and various hematological malignancies [7,10,11,12,13]. Abnormal Gal-9 expression is associated with the development, progression, and metastasis of various cancers. Recently, increasing attention has been given to its prognostic value in cancer [5,9,14]. High Gal-9 expression correlates with increased survival of various solid cancers [12,15,16,17], for instance colon cancer [16] and hepatocellular carcinoma [12]. However, several studies have reported opposing results. For example, in patients with renal cell carcinoma, high Gal-9 expression was linked to poor survival [18]. In patients with acute myeloid leukemia (AML) who failed treatment, Gal-9 expression was also significantly elevated [1,19]. The prognostic role of Gal-9 was also studied in chronic lymphocytic leukemia (CLL). A high concentration of Gal-9 in serum was associated with clinical progression and worse prognosis in CLL patients [20,21].

The current study evaluates Gal-9 as a new prognostic factor in CLL. The relation of Gal-9 expression in malignant B-cells and other routinely measured CLL markers, as well as its clinical relevance, are poorly understood. In this study, Gal-9 expression was quantified by RT-qPCR in purified CD19+ B-lymphocytes of CLL patients and correlated with existing clinical data. Furthermore, we took into account that EBV (Epstein–Barr virus) infection may result in an upregulation of Gal-9 expression in B-cells (LCL) [22] which is why we determined the EBV status of CLL patients. This is an important advancement of the current research. It is important to point out that scientific research has been carried out to gain knowledge about the role of EBV in CLL’s etiology [23,24,25,26,27]. Moreover, to our knowledge, this is the first study in which Gal-9 mRNA expression in CLL cells was correlated with the expression of Ki-67 and PCNA—two the most commonly used proliferation markers.

## 2. Materials and Methods

### 2.1. Patients and Samples

One hundred patients were included in this study, diagnosed with CLL according to the International Workshop on Chronic Lymphocytic Leukemia (IWCLL) criteria [28,29]. Peripheral blood (PB) samples were taken from patients who did not have any history of previous treatment. CLL patients were recruited in the Department of Hematooncology and Bone Marrow Transplantation of the Medical University of Lublin (Lublin, Poland). The median follow-up time from diagnosis was 52 months (range, 1–72). The clinical stage was determined according to the Rai classification system [30]. Moreover, the following clinical and biological characteristics were included in the analysis: white blood cell (WBC) count, lymphocyte count, lactate dehydrogenase (LDH), β2-microglobulin (β2M), ZAP-70 expression (defined as positive if expressed by ≥20% of CLL cells), CD38 expression (cut-off 30%), and cytogenetic abnormalities. In Table 1, we summarize the baseline characteristics of this cohort. Control PB samples were obtained from 27 healthy volunteers (HVs). The control group (aged 35–74 years, median, 55 years) contained 11 females and 16 males. The study protocol was approved by the Ethics Committee of the Medical University of Lublin.

### 2.2. PBMCs Isolation

Peripheral blood mononuclear cells (PBMCs) were isolated using Lymphocyte Separation Medium 1077 (PromoCell, Cat No. C-44010, Heidelberg, Germany). After centrifugation (400× *g*; for 20 min), the PBMC interface was carefully removed and washed twice in PBS (Cat No. ECB4004L, Euroclone, Pero, MI, Italy) at 300× *g* for 5 min. PBMCs were used in the following genetic tests.

### 2.3. Purification of DNA and Quantitative Assay for EBV DNA

DNA was extracted from 5 × 10^6^ PBMCs by QIAamp DNA Blood Mini Kit (Cat No. 51104, Qiagen, Hilden, Germany). The BioSpec-nano spectrophotometer (Shimadzu, Kyoto, Japan) was utilized to estimate the concentration and purity of the isolated DNA. GeneProof Epstein–Barr Virus (EBV) PCR Kit (Cat No. EBV/ISEX/100, Brno, Czech Republic) was used for the detection of EBV DNA. A 7300 Real-Time PCR System (Applied Biosystems, Inc., Waltham, MA, USA) was utilized to amplify the target sequence (EBNA-1; EBV nuclear antigen 1 gene). The PCR cycle protocol consisted of 2 min at 37 °C, 10 min at 95 °C, followed by 45 cycles at 95 °C for 5 s, 60 °C for 40 s, and 72 °C for 20 s. All samples were analyzed in duplicate. This method allows linear quantification of 10^1^ to 10^4^ EBV DNA copies (cp) per µL, as stated by the manufacturer. In our study, 95 out of 100 CLL patients were analyzed for the presence of DNA.

### 2.4. CD19+ B-Lymphocytes Isolation

CD19 MicroBeads (Cat No. 130-050-301, Miltenyi Biotec, Bergisch Gladbach, Germany) were used for B-cell isolation from PBMCs. The manufacturer’s protocol was adhered to. The purity of the achieved cells was over 95%, which was assessed by flow cytometry.

### 2.5. RT-qPCR for Gal-9, Ki67, and PCNA

Total RNA was extracted from PBMCs with the use of a QIAamp RNA Blood Mini Kit (Cat No. 52304, Qiagen, Inc., Valencia, CA, USA) following the manufacturer’s protocol. The amount and quantity of RNA were measured by BioSpec-nano spectrophotometer. Then, 10 ng of total RNA was used in a reverse transcription (RT) reaction. Complementary DNA (cDNA) was synthesized with the QuantiTect Reverse Transcription kit (Cat No. 205311) purchased from Qiagen. RT-qPCR was performed on an Applied Biosystems 7300 Real-Time PCR System, using the TaqMan Gene Expression Assays (assay ID: Gal-9 [Hs00247135_m1], Ki-67 [Hs01032437_m1], PCNA [Hs00427214_g1], Thermo Fisher Scientific, Waltham, MA, USA), and the TaqMan Gene Expression Master Mix (Cat No. 4369016) purchased from Thermo Fisher Scientific Inc. The Human GAPD (GAPDH) Endogenous Control (Cat No. 4310884E, Thermo Fisher Scientific) was used as an internal control. The analysis utilized the cycle quantification value (Cq) and presented it as 2^−ΔCq^ according to the protocol described previously [31].

### 2.6. I-FISH Analysis

The in situ hybridization protocol was described previously [32]. The analysis of all PBMC specimens was performed by FISH using commercially available probes for DLEU1, ATM, TP53, and CEP12 (Vysis CLL FISH Probe Kit; Abbott GmbH, Wiesbaden, Germany) [32].

### 2.7. CD38 and ZAP-70 Expression Analysis

Three-color flow cytometry was utilized to assess the expression of CD38 and ZAP-70 in leukemic (CD19+CD5+) cells. The detection was performed according to previously reported methods [33,34]. Monoclonal antibodies anti-CD19 FITC (clone SJ25C1, Cat No. 340409) and anti-CD5 PE-Cy5 (clone UCHT2, Cat No. 555354) were used to label leukemic B-cells. Furthermore, anti-CD38 PE (clone HIT2, Cat No. 555460) or anti-ZAP-70 PE (clone 1E7.2, Cat No. 344636) (BD Biosciences Franklin Lakes, NJ, USA) were employed to identify CD19+CD5+ cells that expressed CD38 or ZAP-70.

### 2.8. Statistical Analysis

We utilized Statistica 13 PL (StatSoft, Cracow, Poland) and GraphPad Prism 9 (GraphPad Software, San Diego, CA, USA) to perform the data analyses. The difference between the groups was analyzed by the Mann–Whitney U or Kruskal–Wallis tests with post hoc Dunn’s test. Spearman’s rank correlation coefficient (r) was applied to measure the strength of the association between two ranked variables. Kaplan–Meier survival curves were utilized to compare TTT (time to treatment) and overall survival (OS) for Gal-9^low^ and Gal-9^high^ groups. The log-rank test was employed to compare the distribution of time until the occurrence of death or initial treatment in independent groups. Univariate and multivariate-adjusted hazard ratios (HRs) were calculated using Cox proportional hazards models. Receiver operating characteristics (ROC) analysis was conducted to define the optimal cut-off point for the expression of mRNA Gal-9 in leukemic B-cells. Data are expressed as the median and IQR (interquartile range).

## 3. Results

### 3.1. Upregulation of mRNA Gal-9 Expression in Malignant B-Cells from CLL Patients

The expression of Gal-9 mRNA, measured as 2^-ΔCq^, was significantly lower in B-cells obtained from healthy individuals (median (IQR), 2.266 (1.462–3.467)) than in malignant B-cells of CLL patients (median (IQR), 3.616 (2.410–4.924)) (*p* < 0.01) (Figure 1a). The study showed that Gal-9 mRNA expression in CLL patients at Rai 0 (median (IQR), 3.595 (2.266–4.551)) was lower than in those at stages III/IV (median (IQR), 4.370 (2.987–10.05)) (*p* < 0.001) (Figure 1b). The relative expression of Gal-9 mRNA in the patients classified as being at high risk (stage III/IV) of CLL progression was also significantly higher than in intermediate-risk (stage I/II) (IQR), 3.652 (2.406–4.79)) (*p* < 0.01) (Figure 1b).

In addition, CLL patients with a poor prognosis due to high ZAP-70 expression had significantly higher levels of Gal-9 mRNA expression (median (IQR), 4.330 (2.460–7.340)) than those deemed ZAP-70-negative (median (IQR), 3.399 (2.391–4.366) (*p* < 0.05)) (Figure 2a). Conversely, the expression level of Gal-9 mRNA in CD38-negative patients (median (IQR), 3.651 (2.409–4.551)) did not differ significantly (*p* > 0.05) from that in the CD38-positive group (median (IQR), 3.357 (2.415–5.330)) (Figure 2b). In addition, CLL patients with mutated IGHV genes (M-CLL) display lower Gal-9 mRNA expression in purified CD19+ cells (median (IQR), 2.435 (1.891–3.616)) than those with unmutated IGHV genes (U-CLL) (median (IQR), 3.640 (2.409–5.680)%) (*p* < 0.05) (Figure 2c).

In a group of CLL patients, there was a slight positive correlation between the level of Gal-9 mRNA in malignant B-cells and lymphocyte counts (r = 0.224; *p* < 0.05). Despite our observations, there was no significant correlation between Gal-9 mRNA expression and white blood cell counts (WBCs). Likewise, regarding the other laboratory parameters, i.e., serum LDH and β2-microglobulin levels, no significant correlations were detected.

### 3.2. Gal-9 mRNA Relative Expression and Cytogenetic Abnormalities

In the group of patients with 11q-, +12, and/or 17p-, we observed significantly (*p* < 0.001) higher Gal-9 mRNA expression in malignant B-cells vs. group without these unfavorable abnormalities (median (IQR), 4.682 (3.254–7.399) vs. 2.486 (1.903–3.804)), (Figure 3a). In this study, 40 CLL patients had 13q- detected as a sole aberration by FISH. Sole 13q- is associated with a more promising prognosis and better clinical outcomes compared to patients with 17p-, 11q-, or +12 (in isolation or in conjunction with 13q deletion) [35]. In our study, isolated 13q- was not associated with a Gal-9 mRNA expression (Figure 3a). On the other hand, we found a significant (*p* < 0.01) difference in Gal-9 mRNA expression between biallelic and monoallelic del(13q14) types. Patients with heterozygous 13q- had a lower level of Gal-9 mRNA in malignant B-cells compared to homozygous deletion (median (IQR), 3.640 (2.287–4.356) vs. 5.337 (4.058–7.769)) (Figure 3b).

### 3.3. Gal-9 mRNA Expression in EBV-Positive CLL Cases

The relative expression of Gal-9 mRNA in EBV-positive CLL patients (median (IQR), 4.330 (3.031–5.907) was significantly elevated (*p* < 0.05) as compared to EBV-negative ones (median (IQR), 3.335 (2.384–4.379), (Figure 4).

### 3.4. The Clinical Findings and Gal-9 mRNA Expression of CLL Patients

Treatment was required by 49 out of 100 patients during the observation period. The following agents were used for the first-line therapy of CLL: fludarabine + cyclophosphamide + rituximab (n = 14), bendamustine + rituximab (n = 11), chlorambucil + obinutuzumab (n = 10), venetoclax + obinutuzumab (n = 3), rituximab + cyclophosphamide + dexamethasone (n = 4), ibrutinib (n = 4), acalabrutinib (n = 3). The patients who required treatment had a significantly higher expression of Gal-9 mRNA in their leukemic B-cells, as measured at the time of diagnosis (median (IQR), 4.171 (2.440–5.330) vs. 3.576 (2.125–4.342), *p* < 0.05) (Figure 5a). The outcome of treatment was complete remission (CR) in 7 CLL patients (14.29%) and partial remission (PR) in 27 patients (55.10%). The presence of stable disease (SD) was observed in 8 patients (16.32%). The progression of disease (PD) occurred in 70 patients (14.29%). The expression of Gal-9 mRNA was significantly higher in CLL patients with PD (median (IQR), 5.177 (2.871–17.54)) than in those with CR and PR (median (IQR), 2.793 (2.093–4.743)), (*p* < 0.05) (Figure 5b). There were no significant differences in Gal-9 relative expression between patients with CR/PR and those with SD (median (IQR), 3.357 (2.931–4.396)). During the observation period, CLL-related deaths were recorded. The expression of Gal-9 mRNA was significantly higher in patients who passed away (median (IQR), 7.213 (3.878–9.373)) than those who had a complete or partial response (*p* < 0.05) (Figure 5b).

The optimal cut-off value of Gal-9 mRNA in CLL cells was determined using ROC analysis to distinguish between ZAP-70+ and ZAP-70− groups. The optimal threshold for the Gal-9 mRNA expression level was ≥3.389 (AUC, 0.831; 95% CI, 0.748–0.915; *p* < 0.0001; Figure 6a). Next, the group of 100 CLL patients was split into Gal-9^high^ (>3.389; n = 48) and Gal-9^low^ (<3.389; n = 52) groups.

The time from diagnosis to treatment start is a practical endpoint parameter [36]. Time to treatment (TTT) was calculated as the period (in months) between the date of CLL diagnosis and the initiation of the first-line treatment for CLL. In our study, the relative expression of Gal-9 mRNA above 3.389 had a significant influence on TTT (hazard ratio [HR] 2.162; 95% CI 1.071–4.361; *p* < 0.05) (Figure 6b, Table 2). Patients with <3.389 and ≥3.389 levels of Gal-9 mRNA had a median TTT of 51 months and 44 months, respectively. In a multivariate analysis, when considering age, β2M, ZAP-70, and CD38 expression, cytogenetic aberrations, and the presence of EBV-DNA, high Gal-9 mRNA expression was still an independent marker that predicted short TTT for CLL patients (Table 2). However, the expression levels of Gal-9 mRNA in leukemic B-cells did not seem to affect OS (overall survival) (hazard ratio [HR] 1.426; 95% CI 0.625–3.252; *p* < 0.05).

### 3.5. Relationship between Gal-9 and Proliferation Markers (Ki67 and PCNA) mRNA Expression

Gal-9′s role in cell proliferation regulation was considered in this study [22]. As mentioned above, there was a slight positive correlation between lymphocyte count and the expression of Gal-9 mRNA (r = 0.224; *p* < 0.05). The expression levels of PCNA (proliferating cell nuclear antigen) and Ki-67 (determined by RT-qPCR), which are the most frequently used proliferation markers, were correlated with Gal-9 mRNA expression. In the group of CLL patients, there was a moderate positive correlation between the Gal-9 mRNA on one hand and Ki-67 (r = 0.481; *p* < 0.01, Figure 7a) or PCNA mRNA expression on the other (r = 0.551; *p* < 0.01, Figure 7b).

In the Gal-9^low^ group, we observed significantly lower Ki-67 mRNA expression than in the Gal-9^high^ group (median (IQR), 1.921 (0.588–3.556) vs. 3.811 (1.629–9.300), respectively, *p* < 0.001; Figure 8a). Likewise, the PCNA mRNA expression was significantly higher in the Gal-9^high^ group (median (IQR), 2.616 (1.228–5.327)) compared with Gal-9^low^ one (median (IQR), 6.443 (3.546–9.910)%, *p* < 0.0001; Figure 8b).

Taking into account the consistency of high Gal-9 mRNA expression with the presence of EBV DNA for further statistical analysis, the group of 92 CLL patients was divided into four groups: EBV(−)Gal-9^low^ (n = 32), EBV(+)Gal-9^high^ (n = 21), EBV(+)Gal-9^low^ (n = 12), and EBV(−)Gal-9^high^ (n = 27). The combination of Gal-9 mRNA and EBV DNA increased the power of both factors. The EBV(+)Gal-9^high^ group showed significantly higher expression of proliferation markers Ki-67 (Figure 9a) and PCNA (Figure 9b) compared to the EBV(−)Gal-9^low^ and EBV(+)Gal-9^low^ groups.

## 4. Discussion

Understanding the pathomechanisms associated with the clinical and therapeutic heterogeneity of CLL continues to be a challenge for modern personalized medicine. Within a decade, there have been many reports highlighting the role of the microenvironment in the pathogenesis of CLL [37,38,39,40,41]. Immune checkpoints are just one of many components that play an important role in the interactions between leukemic and surrounding cells [42]. Recent years have brought a rapidly increasing number of clinical and preclinical studies of the immune checkpoint blockade (ICB) in CLL patients [42]. Unfortunately, the ICB strategy, especially when it is used as a monotherapy, lacks satisfactory results in CLL. This is likely due to the influence of the immunosuppressive microenvironment. It is, therefore, important to understand the potential prognostic role of checkpoint molecules in the pathogenesis of CLL in relation to other prognostic factors, disease stage, and the prospect of achieving complete remission. One of the key immune checkpoints, whose role in the pathogenesis of CLL is not fully understood, is Gal-9 [2]. In the current study, we report the expression level of Gal-9 in purified CD19+ B-lymphocytes of CLL patients.

Our data are concordant with those reported by Taghiloo et al. [13] who showed significantly higher Gal-9 expression in CLL-derived PBMCs than in healthy controls. It is worth mentioning that we verified their outcomes in a larger group of 100 CLL patients. Llaó Cid et al. [43], using a single-cell RNA-sequencing analysis, identified CLL cells and dendritic cells as a source of Gal-9. They demonstrated an increase in *LGALS9* gene expression in CLL.

Studies by Wdowiak et al. [11], Pang et al. [44], and Alimu et al. [20] found increased serum Gal-9 levels in CLL patients. However, one has to remember that Gal-9 is known to be produced and secreted by various cell types including cancer cells and neighboring non-neoplastic cells (T- and B-lymphocytes, monocytes, dendritic cells, macrophages, etc.) [1]. Additionally, Gal-9 can be found both extra- and intracellulary. In the latter case Gal-9 localizes most commonly in cytoplasm or on cell surface [3,5,45]. Gal-9′s function at least partially depends on its location, i.e., intracellularly it modulates signal transduction, proliferation, differentiation, and apoptosis, while extracellularly Gal-9 regulates adhesion, chemotaxis, and local inflammation. The impact of Gal-9 on tumor progression remains unclear [3,46]. Several studies so far have focused on Gal-9 in cancer patients [5,9,14], including CLL patients [11,13,20]. Galectin-9′s actual role in CLL has yet to be fully understood. It is known that through its interaction with TIM-3, Galectin-9 regulates the survival of tumor cells [5]. Gal-9 and TIM-3′s binding results in the exhaustion or apoptosis of effector T-cells [5,13]. Recent studies showed that TIM-3 expression is increased in CD4+ and CD8+ T-cells in CLL patients [20,47,48]. It was also revealed that the Gal-9/Tim-3 pathway promoted the function of Treg cells that are involved in the immune escape of CLL [44]. Additionally, Alimu X et al. [20] found that increased levels of Gal-9 in the serum of CLL patients correlated with a higher percentage of MDSC (myeloid-derived suppressor cells) in the patient’s blood and disease progression. This study confirms that Gal-9 is an important player in forming an immunosuppressive microenvironment [20,49].

According to Bozorgmehr et al. [50], shedding of Gal-9 from leukemic B-cells leads to an increase in plasma Gal-9 in CLL patients. The authors found that Gal-9 in CLL patients contributed to increased apoptosis of polyfunctional CD26^high^CD8+ T-cells [50]. Another study (presented during the 2022 ASH Annual Meeting) revealed that treating mice with Gal-9 blocking antibodies slowed the development of the disease in an Eµ-TCL1-induced mouse model of CLL [43]. Therefore, high Gal-9 expression can be used by CLL cells to escape immune surveillance.

In our study, the expression of Gal-9 was significantly higher in the CLL group with a high risk (stage III/IV) of progression than in low (stage 0) or intermediate risk (stage I/II), which was consistent with Taghillo et al. [13]. The investigators [13] evaluated Gal-9 mRNA expression in mononuclear cells of 25 CLL patients at various clinical stages. Our study evaluated Gal-9 mRNA expression in isolated CD19+ B-cells from 100 patients with newly diagnosed CLL. Alimu et al. [20] observed that Gal-9 levels in CLL patients increased with the Binet stage, and similar results were obtained by Wdowiak et al. [11].

Classical Rai and Binet staging systems cannot identify patients with poor prognosis because most patients are diagnosed at the early stages. Thus, other markers, such as mutation status of IGHV, ZAP-70 and CD38 expression, cytogenetic abnormalities, or serologic markers (β2M and LDH) have become important prognostic factors for CLL [51,52,53]. IGHV gene mutational status is a powerful indicator of prognosis in CLL [54]. However, routine analysis of IGHV is a challenging and costly task for most clinical laboratories. ZAP-70 expression in CLL cells, which can be detected through flow cytometry, is associated with IGHV mutational status, survival, and disease progression [52,55,56].

Our study is the first to evaluate Gal-9 expression in CLL patients differentiated by ZAP-70, CD38, and IGHV status. Significantly higher expression of Gal-9 was observed in ZAP-70-positive patients and CLL cases with unmutated IGHV. The potential prognostic value of Gal-9 in CLL cells was also evaluated by analyzing the association between Gal-9 mRNA level and cytogenetic abnormalities, serum β2M, and LDH levels. In our study, no association between Gal-9 mRNA level, white blood cell count, β2M and CD38 expression was found. In this respect, it should be noted that at first, CD38 expression was suggested as a substitute indicator of IGHV mutation status due to high levels of CD38 expression in CLL cells with unmutated IGHV genes. However, there is no absolute association between CD38 expression and IGHV mutational status. Moreover, the expression of CD38 may change as the disease progresses [53]. This may cause variable results of CD38 expression assessment as opposed to those obtained for ZAP-70 expression and IGHV mutation status analysis.

Due to the heterogeneous clinical picture of CLL, it is very important to divide patients into appropriate risk groups in the context of detected chromosomal aberrations [11]. It must be emphasized that higher Gal-9 expression in malignant B-cells was observed in patients with deletion 17p13.1, deletion 11q22.3, and/or trisomy 12 compared with patients without adverse cytogenetic aberrations. This is in agreement with Wdowiak et al. [11] who reported that CLL patients with del(11q) and del(17p) showed higher Gal-9 serum levels than patients without those adverse aberrations.

Deletion of 13q14 is the most prevalent cytogenetic abnormality in CLL. It is widely considered as a positive prognostic factor if it is a sole aberration. Despite that, the clinical course of CLL cases with 13q- is diverse. Some patients exhibit a more aggressive clinical course and die rapidly, even after starting treatment at the time of diagnosis [35,57]. This is the reason why 13q14 deletions should be further classified as monoallelic or biallelic. A more aggressive CLL course is linked to the presence of homozygous del(13q14) [35]. Studies have indicated that the biallelic 13q14 deletion is a negative prognostic factor that is associated with faster lymphocyte growth and inferior prognosis [35,58,59]. Indeed, we observed a lower expression of Gal-9 in patients with monoallelic 13q14 compared to biallelic deletion.

Patients who needed therapy had significantly higher Gal-9 mRNA levels at the time of diagnosis compared to those who did not receive treatment during the observation period. Furthermore, our research shows that Gal-9 expression is lower in CLL patients with CR and/or PR compared to those with disease progression. The association between high Gal-9 and a shorter time to treatment was established. A shorter TTT was observed in CLL patients with high Gal-9 expression. In the multivariate analysis, baseline Gal-9 held its independent prognostic factor status for the TTT. This is an important and new implication of the current study. Alimu et al. [20] and Wdowiak et al. [11] found that Gal-9 serum levels in CLL patients increased with the progression of the Binet stage. In addition, the ROC analysis results of Alimu et al. [20] showed that soluble Gal-9 had relatively high sensitivity and specificity in evaluating disease progression. The findings revealed that patients with high Gal-9 had a poor prognosis [20]. We suggest that Gal-9 levels in malignant B-cells may be used as a novel prognostic marker for CLL patients. In addition, for the first time, we have combined the expression level of Gal-9 mRNA with such a broad clinical context.

In the past, it was believed that defective apoptosis rather than proliferation caused the slow accumulation of CLL cells in vivo [60,61]. Nonetheless, this viewpoint has evolved. CLL cells in vivo were found to have significant continuous proliferation [61]. Moreover, proliferation increases with disease progression [60,61,62,63]. It should also be noted that malignant B-cells depend on survival signals that they receive from neighboring non-neoplastic cells [64]. As mentioned above, we found a positive correlation between lymphocyte count and Gal-9 expression in leukemic B-cells. Therefore, we took into consideration the fact that Gal-9 is engaged in the regulation of cell proliferation [22]. The expression of Gal-9 mRNA was correlated with that of Ki-67 and PCNA, two of the most commonly used proliferation markers [65]. The results of our study demonstrated an association between the Gal-9 mRNA expression in CLL cells and Ki-67 and PCNA mRNA expression. In the study by Giglio et al. [66], PCNA had higher expression in CLL samples than in normal PBMCs. Moreover, Giglio et al. [67] found that PCNA level was linked to cell proliferation, clinical stages, and lymphocyte doubling time [67]. Moreover, high levels of PCNA expression by unstimulated CLL lymphocytes at the time of diagnosis may identify patients with higher proliferative activity and, therefore, with a worse prognosis. Šoljić V et al. [68], in a study performed on native bone marrow and peripheral blood smears from 65 CLL patients, detected higher Ki-67 expression in ZAP-70-positive patients. It is worth recalling that we found an association between the expression of ZAP-70 in CLL cells (i.e., ZAP-70 level ≥ 20%) and high Gal-9 expression. High ZAP-70, Gal-9, and proliferation markers’ expression suggest poorer prognosis in CLL cases. The study by Bruey et al. [61], found that high levels of circulating Ki-67 in plasma (cKi-67) were linked to shorter survival.

According to Xu et al. [22], EBV infection may induce Gal-9 expression in B-cells. Proliferation and growth of lymphoblastoid cell lines (LCL) is triggered when peripheral blood B-cells are infected with EBV in vitro [69]. The majority of EBV-associated cancers are lymphomas that originate from B-cells (e.g., Burkitt lymphoma [70] and diffuse large B-cell lymphomas (DLBCLs) [69,71,72]) as well as carcinomas derived from epithelial cells (e.g., nasopharyngeal carcinoma and gastric carcinoma) [69]. For many years, scientific research has also been conducted to gain knowledge about the place of EBV in the etiology of CLL [26,27,73]. The study of Visco et al. [27] showed that EBV DNA load in patients with CLL at diagnosis had a strong relationship with overall survival. Furthermore, the increase in EBV DNA load was directly linked to adverse outcomes. It was also found that EBV-DNA-positive subjects were also more likely to have Richter transformation [73]. Using an in vitro EBV transformation of B-cells as an experimental model, Xu et al. [22], observed a sustained increase in Gal-9 mRNA expression. By blocking or forcing Gal-9 expression in the early stage of EBV infection, they gained evidence that Gal-9 facilitated the establishment of latent infection and the outgrowth of immortalized clones. Our study revealed significantly increased Gal-9 mRNA levels in EBV-positive CLL patients in comparison to EBV-negative ones. Additionally, taking into account the consistency of high Gal-9 mRNA expression with the presence of EBV DNA, CLL patients were divided into four groups: EBV(−)Gal-9^low^, EBV(+)Gal-9^high^, EBV(+)Gal-9^low^, and EBV(−)Gal-9^high^. The EBV(+)Gal-9^high^ group showed the highest expression of proliferation markers Ki-67 and PCNA in comparison with the other groups.

## 5. Conclusions

The biological role of Gal-9 in CLL cannot be fully defined based on our observations, as we acknowledge; more research is required. However, our results suggest that changes in Gal-9 mRNA expression in malignant B-cells may become a useful clinical tool to monitor disease progression and determine treatment outcome. Our proposal is that EBV coinfection can worsen the prognosis of CLL patients, partly due to Gal-9 upregulation caused by EBV.

## Figures and Tables

**Figure 1 cancers-15-05370-f001:**
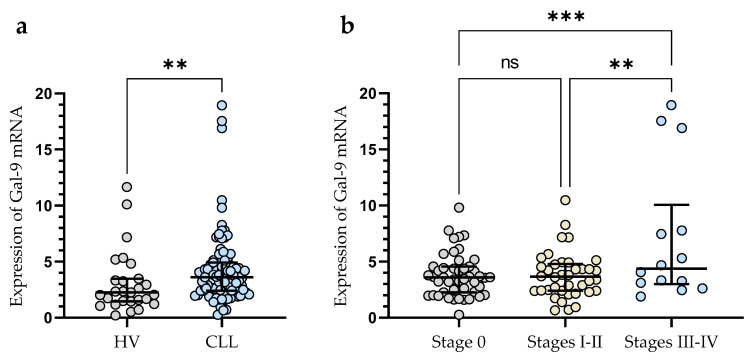
Level of Gal-9 mRNA in CLL cells and B-lymphocytes from healthy volunteers (HV) (**a**). Gal-9 mRNA expression at various Rai stages (**b**). The median is represented through the central line. “Whiskers” indicate the IQR (interquartile range). ** *p* < 0.01, *** *p* < 0.001; ns, not significant.

**Figure 2 cancers-15-05370-f002:**
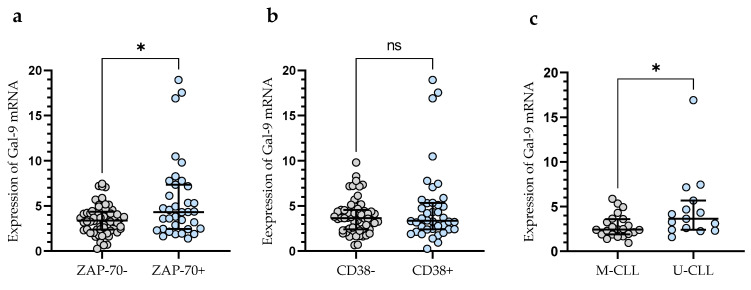
Gal-9 mRNA expression in malignant B-cells from ZAP-70-positive and ZAP-70-negative (**a**) CLL patients and from CD38-positive and CD38-negative groups (**b**). CLL patients with different IGHV mutation status (**c**). The individual values and median with IQR (interquartile range) are displayed. * *p* < 0.05; ns, not significant. IGHV, immunoglobulin heavy chain variable gene; M-CLL, mutated IGHV; U-CLL, unmutated IGHV; ZAP-70, zeta-chain-associated protein kinase 70.

**Figure 3 cancers-15-05370-f003:**
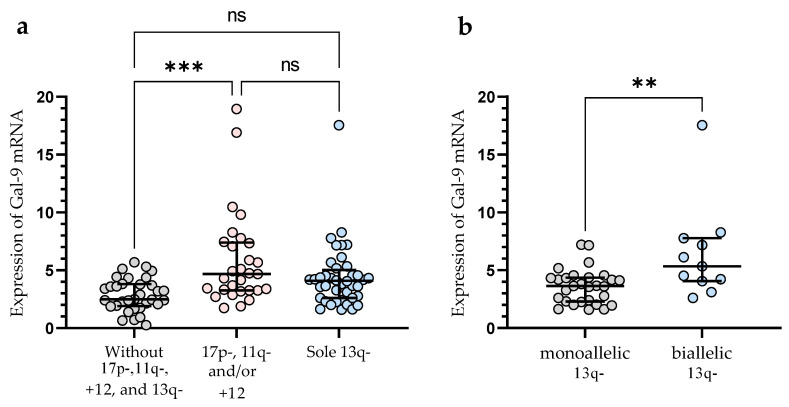
Level of Gal-9 mRNA in patients with cytogenetic changes (17p-, 11q-, or +12 in isolation or in conjunction with 13q deletion) (**a**). Gal-9 mRNA expression in a group of CLL patients with sole 13q deletion type (**b**). The individual values and median with IQR are displayed. ** *p* < 0.01, *** *p* < 0.001; ns, not significant; IQR, interquartile range.

**Figure 4 cancers-15-05370-f004:**
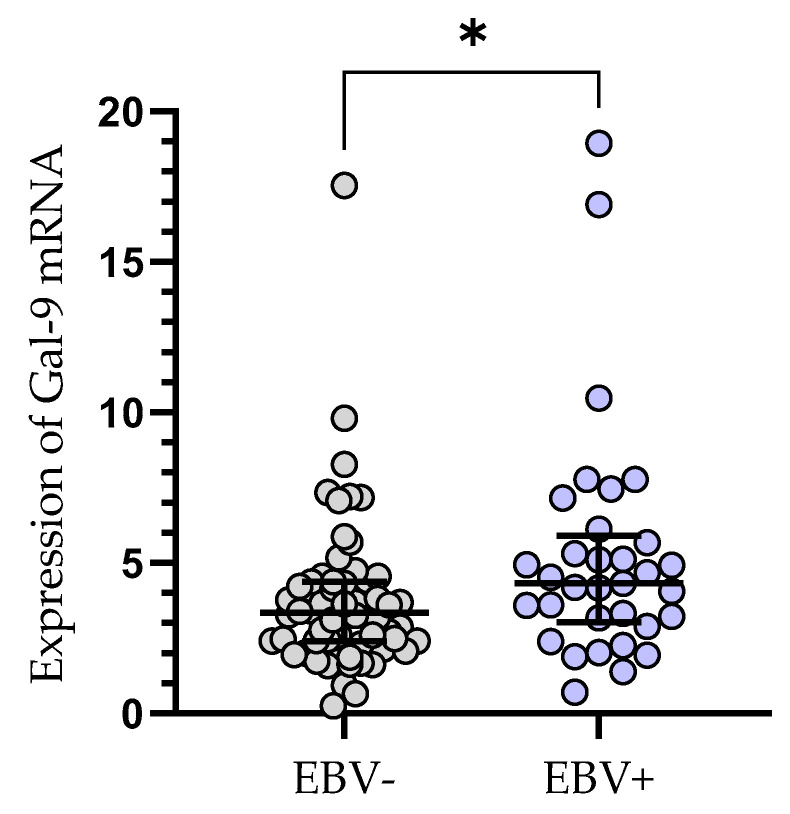
Gal-9 mRNA expression in leukemic B-cells in EBV-negative (n = 62) and EBV-positive (n = 33) CLL patients. Individual values and median with IQR are shown. * *p* < 0.05. EBV, Epstein–Barr virus; IQR, interquartile range.

**Figure 5 cancers-15-05370-f005:**
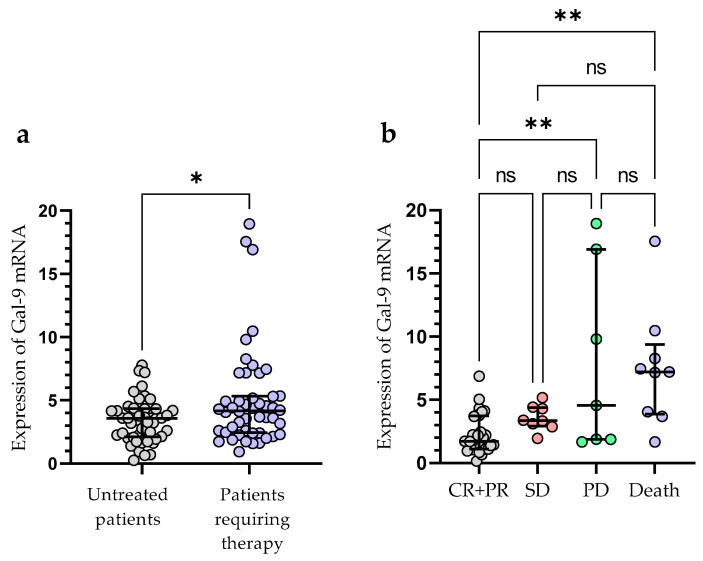
Patients with CLL who needed therapy and those who did not require therapy were compared according to Gal-9 mRNA expression levels (**a**). The expression levels of Gal-9 mRNA in CLL patients who respond to treatment and those with progressive disease (**b**). The individual values and median with IQR are displayed. ** *p* < 0.01, * *p* < 0.05; ns, not significant; CR, complete response; PR, partial response; SD, stable disease; PD, disease progression; IQR, interquartile range.

**Figure 6 cancers-15-05370-f006:**
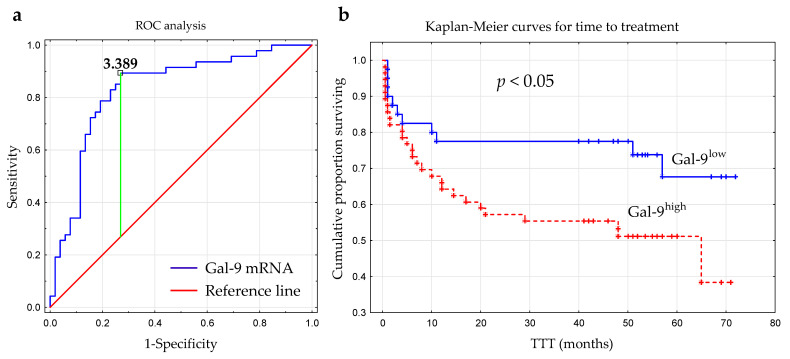
Receiver operating characteristic curve (ROC) was used to determine the optimal cut-off for Gal-9 mRNA expression in malignant B-cells. The calculation included the cut-off value (3.389), AUC (0.831), sensitivity (89%), specificity (73%), and Youden index (0.62) (**a**). Kaplan–Meier survival curves for Gal-9^low^ (<3.389) and Gal-9^high^ (≥3.389) groups comparing TTT (**b**). TTT, time to treatment; AUC, area under the ROC curve.

**Figure 7 cancers-15-05370-f007:**
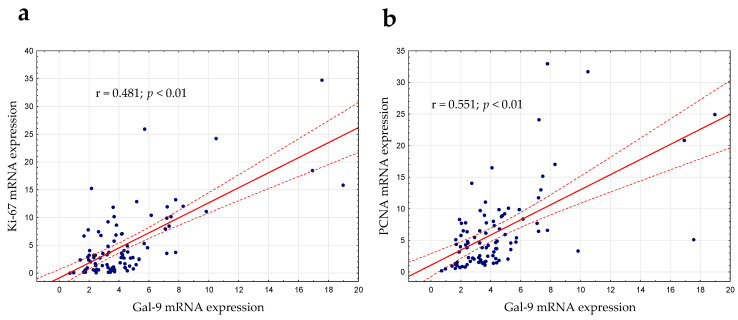
Ki-67 (**a**) and PCNA (**b**) mRNA expressions were correlated with Gal-9 mRNA expression determined by RT-qPCR. The strength of the association between variables was measured using Spearman’s rank correlation coefficient (r). PCNA, proliferating cell nuclear antigen.

**Figure 8 cancers-15-05370-f008:**
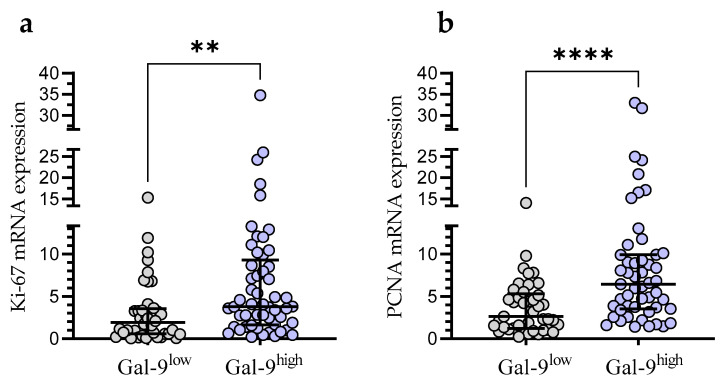
Ki-67 (**a**) and PCNA (**b**) mRNA expressions in Gal-9^low^ (<3.389) and Gal-9^high^ groups (≥3.389). Individual values and median with IQR are shown. ** *p* < 0.001, **** *p* < 0.0001. PCNA, proliferating cell nuclear antigen; IQR, interquartile range.

**Figure 9 cancers-15-05370-f009:**
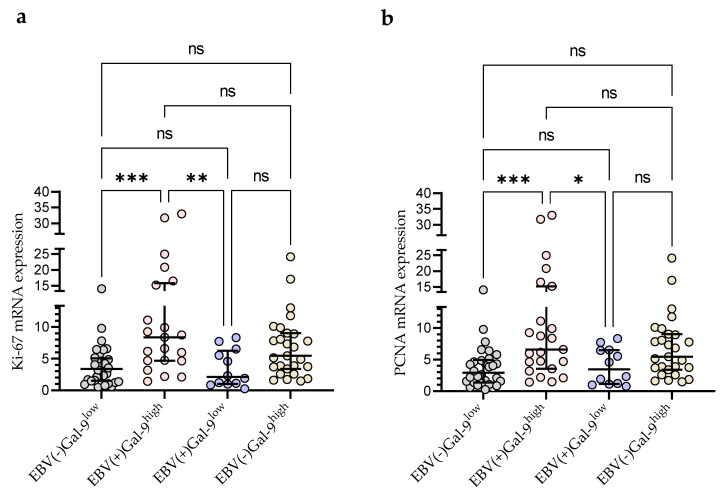
Comparison of Ki-67 (**a**) and PCNA (**b**) mRNA expression between four groups: EBV(−)Gal-9^low^ (n = 32), EBV(+)Gal-9^high^ (n = 21), EBV(+)Gal-9^low^ (n = 12), EBV(−)Gal-9^high^ (n = 27). Individual values and median with IQR are shown. * *p* < 0.05, ** *p* < 0.01, *** *p* < 0.0001, ns, not significant. PCNA, proliferating cell nuclear antigen; IQR, interquartile range.

**Table 1 cancers-15-05370-t001:** Baseline characteristics of the CLL patient cohort.

Characteristics	n
Total number of patients	100
Rai Stage 0	46
Rai Stages I–II	40
Rai Stages III–IV	14
ZAP-70	
≥20%	35
<20%	65
CD38	
≥30%	39
<30%	61
FISH	
17p-	5
11q-	15
+12	7
sole 13q-	40
Heterozygous 13q-	28/40
Homozygous 13q-	11/40
Without 17p-, 11q-, +12, and 13q-	33
Patients treated during observation period	49
Complete remission (CR)	7/49
Partial remission (PR)	27/49
Stable disease (SD)	8/49
Disease progression (PD)	7/49
Untreated patients	51
The IGHV mutational status	
Unmutated IGHV	15
Mutated IGHV	22
Not available	63
EBV(−)	62
EBV(+)	33
not available	5
Age in years at diagnosis, median (range)	67 (38–85)
White blood cells (WBC) count (G/L), median (IQR)	25.84 (17.28–54.80)
Lymphocyte count (G/L), median (IQR)	18.30 (10.68–44.96)
Lactate dehydrogenase (LDH) level (IU/L), median (IQR)	379 (324–428)
β2-microglobulin level (mg/dL), median (IQR)	2.45 (2.01–3.18)
ZAP-70-positive CD19+/CD5+ cells (%), median (IQR)	7.42 (3.67–20.10)
CD38-positive CD19+/CD5+ cells (%), median (IQR)	6.72 (1.595–34.23)

IQR, interquartile range; EBV, Epstein–Barr virus; IGHV, immunoglobulin heavy chain variable gene; ZAP-70, zeta-chain-associated protein kinase 70.

**Table 2 cancers-15-05370-t002:** Cox regression analysis for TTT in CLL patients.

Risk Factors	Univariate Analysis			Multivariate Analysis		
	HR	95% CI	*p*-Value	HR	95% CI	*p*-Value
Age						
≥65 years	1.95	0.971–3.936	ns	NA		
<65 years						
ZAP-70						
≥20%	1.587	0.838–3.008	<0.01	1.697	0.506–5.695	<0.05
<20%						
CD38						
≥30%	1.785	0.952–3.350	ns	NA		
<30%						
β2M						
≥3.5 mg/dL	2.293	1.172–4.488	<0.01	2.734	1.096–6.821	<0.05
<3.5 mg/dL						
17p-, 11q- or +12						
Positive	2.03	1.051–3.923	<0.01	2.327	0.980–5.529	<0.05
Negative						
Sole 13q-						
Positive	0.482	0.249–0.930	<0.05	0.766	0.300–1.953	ns
Negative						
EBV-DNA						
Positive	1.366	0.695–2.687	ns	NA		
Negative						
Gal-9 mRNA						
≥3.389	2.162	1.071–4.361	<0.05	0.472	0.232–0.956	<0.05
<3.389						

The multivariate analysis was performed with variables that had *p* < 0.05 in the univariate analysis. β2M, β2 microglobulin; HR, hazard ratio; 95% CI: 95% confidence interval. NA, not assessed; ns, not significant; EBV, Epstein–Barr virus.

## Data Availability

The data presented in this study are available within the article. Other data that support the findings of this study are available upon request from the corresponding author.

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
