# Peer review of "Prognostic Potential of Galectin-9 mRNA Expression in Chronic Lymphocytic Leukemia"

_cancers, 2023, doi:10.3390/cancers15225370_

Round 1

Reviewer 1 Report

Comments and Suggestions for Authors

In their submitted manuscript, the authors analyzed the expression of galectin-9 by RT-qPCR in immunomagnetically isolated B cells from 100 patients with chronic lymphocytic leukemia (CLL). They found that galectin-9 mRNA is upregulated in CLL patients in comparison with healthy controls. Increased expression of galectin-9 mRNA was observed in patients at Rai III/IV stages, ZAP-70-positive patients, patients with adverse cytogenetics (del17p, del11q and/or trisomy 12) and biallelic del13q, EBV+ patients, therapy requiring patients and those with progressive disease after treatment. The expression of galectin-9 also correlated with absolute lymphocyte counts and proliferation markers (Ki67 and PCNA) mRNA expression. Finally, high galectin-9 mRNA expression was associated with significantly shorter time to therapy in CLL patients.

I have the following comments and questions:

1. What was the purity of B cells after their immunomagnetic isolation using CD19 microbeads. Did you analyze it, e.g. by flow cytometry?

2. Galectin-9 mRNA expression was higher in ZAP-70-positive but not in CD38-positive patients. Both ZAP-70 and CD38 gained a lot of attention in the past as potential surrogate markers for IgHV unmutated status. It would be interesting to see if there is a direct relationship between galectin-9 mRNA expression and IgHV mutational status.

3. IgHV mutational status is an important prognostic factor included in the CLL-International prognostic index (CLL-IPI). Did you correlate galectin-9 mRNA expression with CLL-IPI?

4. Forty-nine out of 100 CLL patients required therapy. Unfortunately, the authors do not specify what kind of treatment these patients received.

5. In relation to the previous point, the outcome of treatment in 49 patients was: 10.2% CR, 22.4% PR, 26.5% SD and 40.8% PD. I think this is a surprisingly poor response rate considering all these 49 patients were therapy-naive. This discrepancy should be addressed by the authors and adequate information on patient treatments should be provided.

Comments on the Quality of English Language

Only a few minor corrections are required.

Author Response

Dear Reviewer 1, thank you very much for your valuable suggestions and comments. Indeed, in its past form, our manuscript had several weaknesses. We improved our work according to your remarks. We believe that this paper will meet your expectations. Please find the more detailed answers to your suggestions below.

Point 1. What was the purity of B cells after their immunomagnetic isolation using CD19 microbeads. Did you analyze it, e.g. by flow cytometry?

Response 1. In the Materials and Method section, details about the purity of isolated B cells were added (section 2.4, lines 118-119).

Point 2. Galectin-9 mRNA expression was higher in ZAP-70-positive but not in CD38-positive patients. Both ZAP-70 and CD38 gained a lot of attention in the past as potential surrogate markers for IgHV unmutated status. It would be interesting to see if there is a direct relationship between galectin-9 mRNA expression and IgHV mutational status.

Response 2. IGHV mutation status was available only in 37 out of 100 CLL patients (Table 1). Nonetheless, in a revised version of our manuscript, the comparison of Gal-9 mRNA expression between mutated-IGHV and unmutated-IGHV groups was added (lines: 181-191). Figure 2 was changed, (C) panel was added. In our study Gal-9 mRNA expression was higher in ZAP-70-positive but not in CD38-positive patients. These discrepancies were explained in the discussion (lines 385-391).

Point 3. IgHV mutational status is an important prognostic factor included in the CLL-International prognostic index (CLL-IPI). Did you correlate galectin-9 mRNA expression with CLL-IPI?

Response 3. CLL-IPI combines 5 parameters: age, clinical stage, TP53 status [normal vs. del(17p) and/or TP53 mutation], IGHV mutational status, and serum β2-microglobulin. Unfortunately, IGHV mutation status was available only in 37 CLL patients. IGHV mutation testing is not routinely performed in our laboratory. So we couldn't stratify all patients with CLL into four risk categories. Thank you very much for this suggestion. In the future, we will try to supplement our results with the CLL-IPI index.

Point 4. Forty-nine out of 100 CLL patients required therapy. Unfortunately, the authors do not specify what kind of treatment these patients received.

Response 4. The information on patient treatments was provided in section 3.4. (lines: 225-229).

Point 5. In relation to the previous point, the outcome of treatment in 49 patients was: 10.2% CR, 22.4% PR, 26.5% SD and 40.8% PD. I think this is a surprisingly poor response rate considering all these 49 patients were therapy-naive. This discrepancy should be addressed by the authors and adequate information on patient treatments should be provided.

Response 5. This discrepancy was addressed and corrected (Table 1, lines: 232-238). The information on patient treatments was provided (lines: 225-229). Figure 5b has been changed.

Reviewer 2 Report

Comments and Suggestions for Authors

In this study, the correlation between Gal9 and different parameters of CLL were tested. However, it appears that the role of Gal9 as a prognostic factor in CLL is already known. Therefore the enthusiasm is lowered. No mechanistic details are provided. Could the authors highlight the novelty of the study?

What is the mechanism by which Galectin-9 is upregulated in CLL and what is the role of Gal9 in CLL? Could the authors comment on that?

Can the authors verify the use of Gal9 as a prognosticator in CLL using publicly available data (if such data can be accessed by the authors)?

Minor comments

What is the significance of ZAP-70 and CD38 in CLL? Why were these 2 markers used for stratification?

Please explain what time from diagnosis to treatment start means.

Author Response

Dear Reviewer 3, we appreciate your comments. Indeed, in its past form, our manuscript had several weaknesses. We improved our work according to your remarks. We believe that this paper will meet your expectations. Please find the more detailed answers to your suggestions below

Point 1. In this study, the correlation between Gal9 and different parameters of CLL were tested. However, it appears that the role of Gal9 as a prognostic factor in CLL is already known. Therefore the enthusiasm is lowered. No mechanistic details are provided. Could the authors highlight the novelty of the study?

Response 1. The role of Gal-9 as a prognostic factor in CLL is still investigated because Gal-9 has various isoforms. Recent studies showed that elevated serum levels of the soluble Gal-9, assessed by ELISA, have been identified as a potential prognostic factor in patients with CLL. However, these studies do not identify exactly which cells release the soluble form of Gal-9. For the first time in 2017, Taghiloo et al. observed elevated Gal-9 mRNA expression in mononuclear cells in 25 CLL patients at various clinical stages. Our study is much more groundbreaking compared to previous papers on that topic because it evaluates Gal-9 mRNA expression in isolated CD19+ B cells from 100 patients with newly diagnosed CLL. Moreover, we analyzed the Gal-9 mRNA expression levels with several prognostic clinical and biological factors (e.g. ZAP-70 and CD38 expression, chromosomal aberrations, IGHV mutations). Moreover, we also highlighted the difference in Gal-9 mRNA expression in patients who required CLL treatment during follow-up, including treatment response so we also noted a possible predictive role of Gal-9 mRNA. In addition, for the first time, we have combined the expression level of Gal-9 mRNA with such a broad clinical context. Of note, we are the first team to correlate Gal-9 mRNA expression levels with the proliferation markers (i.e. Ki-67 and PCNA) of CLL cells taking into account the presence or absence of EBV DNA.

The novelty of this study was highlighted in the Introduction (lines 71-76) and Discussion (lines 379-380, 415-416, 421-423).

Point 2. What is the mechanism by which Galectin-9 is upregulated in CLL and what is the role of Gal9 in CLL? Could the authors comment on that?

Response 2. Thank you for pointing out these aspects. We have added comments on the role of Gal-9 in CLL and the mechanism by which Galectin-9 is up-regulated in CLL in the Discussion section (lines 347-363).

Point 3. Can the authors verify the use of Gal9 as a prognosticator in CLL using publicly available data (if such data can be accessed by the authors)?

Response 3. Bioinformatics Available datasets (Broad, Nature 2015, Broad, Cell 2013; ICGC, Nature Genetics 2011; IUOPA) were used to assess the expression of LGALS9 gene in leukemic B cells. Unfortunately, the data sets did not contain LGALS9 expression data.

Minor comments

Point 1. What is the significance of ZAP-70 and CD38 in CLL? Why were these 2 markers used for stratification?

Response 1. In the revised version of our manuscript, we added information about ZAP-70 and CD38 expression significance in CLL in the Discussion: lines 372-378: and lines 387-389.

Point 2. Please explain what time from diagnosis to treatment start means.

Response 1. In our study, the results obtained were analyzed for prognostic factors, clinical data, and endpoints such as time to treatment (TTT) and overall survival time (OS). Time to Treatment (TTT) was calculated as the period (in months) between the date of CLL diagnosis and the initiation of the first-line treatment for CLL. It was added in lines 266-267.

Round 2

Reviewer 2 Report

Comments and Suggestions for Authors

Authors have sufficiently addressed most of my concerns.

One question remains, if GAL9 expression is not seen in published databases, does it mean it is not expressed? Or that the methodologies used were insufficient or its because of the isoform variation?

Author Response

Dear Reviewer 2, we carefully studied your raised questions. Please find the detailed answer below. We believe that this answer will meet your expectations.

One question remains, if GAL9 expression is not seen in published databases, does it mean it is not expressed? Or that the methodologies used were insufficient or its because of the isoform variation?

Response: In the Discussion section (lines: 333-335 and 359-361), we highlighted that during ASH2022, a study with single-cell omic analyses was presented showing that Gal-9 expression on leukemic B cells led to T-cell dysfunction, and that treatment with Gal-9 antibody of a mouse model of CLL reduced disease progression [Llaó Cid L, 2022]. This data is pending acceptance in the European Genome Phenome Archive (EGA) and Gene Expression Omnibus. Therefore, it explains why it is not yet visible in published data. Other available datasets did not contain expression data - those assessed mutations. Broad, Nature 2015 is the only dataset that contains expression levels. Authors used microarrays instead of real NGS. We cannot fully identify the exact microarray used in the study - there is a possibility that it did not recognize LGALS9. The methodology seems to be a limitation.
